# Immune Response to Seasonal Influenza Vaccination in Multiple Sclerosis Patients Receiving Cladribine

**DOI:** 10.3390/cells12091243

**Published:** 2023-04-25

**Authors:** Leoni Rolfes, Steffen Pfeuffer, Jelena Skuljec, Xia He, Chuanxin Su, Sinem-Hilal Oezalp, Marc Pawlitzki, Tobias Ruck, Melanie Korsen, Konstanze Kleinschnitz, Derya Aslan, Tim Hagenacker, Christoph Kleinschnitz, Sven G. Meuth, Refik Pul

**Affiliations:** 1Department of Neurology, HeinrichHeine University Düsseldorf, 40225 Duesseldorf, Germany; 2Department of Neurology, University Hospital Giessen and Marburg, Justus-Liebig-University Giessen, 35392 Giessen, Germany; 3Department of Neurology and Center for Translational Neuro and Behavioral Science, University Medicine Essen, 45127 Essen, Germany; 4Center for Translational Neuro and Behavioral Sciences (C-TNBS), University Medicine Essen, 45127 Essen, Germany

**Keywords:** multiple sclerosis, cladribine, immunization, influenza, vaccination

## Abstract

Cladribine has been approved for the treatment of multiple sclerosis (MS) and its administration results in a long-lasting depletion of lymphocytes. As lymphopenia is known to hamper immune responses to vaccination, we evaluated the immunogenicity of the influenza vaccine in patients undergoing cladribine treatment at different stages vs. controls. The antibody response in 90 cladribine-treated MS patients was prospectively compared with 10 control subjects receiving platform immunotherapy (NCT05019248). Serum samples were collected before and six months after vaccination. Response to vaccination was determined by the hemagglutination-inhibition test. Postvaccination seroprotection rates against influenza A were comparable in cladribine-treated patients and controls (H1N1: 94.4% vs. 100%; H3N2: 92.2% vs. 90.0%). Influenza B response was lower in the cladribine cohort (61.1% vs. 80%). The increase in geometric mean titers was lower in the cladribine group vs. controls (H1N1: +98.5 vs. +188.1; H3N2: +225.3 vs. +300.0; influenza B: +40.0 vs. +78.4); however, titers increased in both groups for all strains. Seroprotection was achieved irrespective of vaccination timing and lymphocyte subset counts at the time of vaccination in the cladribine cohort. To conclude, cladribine-treated MS patients can mount an adequate immune response to influenza independently of treatment duration and time interval to the last cladribine administration.

## 1. Introduction

In the last two decades, the approval of new and effective disease-modifying treatments (DMT) led to a significant change in the therapeutic strategy of multiple sclerosis (MS). New so-called immune reconstitution therapies (IRTs) have the potential to induce long-term or even permanent drug-free remission in people with MS [1]. These therapies deplete components of the immune system, intending to allow its renewal [2]. In this context, cladribine represents the first short-course oral IRT approved for the treatment of active relapsing MS after having been positively evaluated in placebo-controlled randomized clinical trials [3,4]. Cladribine is a synthetic purine analogon that induces lymphocyte depletion due to an accumulation of intracellular chloro-deoxyadenosine triphosphate. Immunophenotyping studies showed that cladribine only modestly affects T cells, whereas it vastly reduces B-cell counts [5], particularly class-switched and unswitched memory B cells [6,7].

Given those immunocompromising effects, the impact of this drug on immunization responses cannot be ruled out. Therefore, it is recommended to complete vaccination requirements as per local prescribing information before initiation of cladribine to optimize vaccine effectiveness. However, protection from some pathogens, such as influenza viruses, requires annually repeated vaccinations.

The immunogenicity of the influenza vaccine in MS patients depends on the immunomodulatory treatment they received. A sufficient vaccination response to influenza has been shown in patients receiving interferon (IFN)-β, glatiramer acetate, dimethyl fumarate, and teriflunomide therapy [8,9,10,11,12,13]. In contrast, a reduced likelihood of seroprotection following influenza vaccination was observed in individuals on fingolimod, natalizumab, or the B-cell-depleting therapy ocrelizumab [9,11,14].

With respect to cladribine, a study of 12 patients demonstrated seroprotection after influenza vaccination in all participants, regardless of the time passed since the last treatment administration or total lymphocyte count [15]. However, the study was retrospective in design, included a limited number of patients, and a control group was not established. Consequently, larger and prospective studies are required to evaluate whether cladribine treatment influences the likelihood of response to influenza immunization.

In this study, we analyzed the immunogenicity of the seasonal influenza vaccine 2020/2021 in patients with relapsing MS receiving immunomodulatory cladribine therapy, with baseline sampling and follow-up after 6 months in both the study population and controls (MS patients treated with platform DMTs).

## 2. Materials and Methods

### 2.1. Study Design

CIRMS (Cladribine on Immune Responses in Multiple Sclerosis) was designed as a large prospective observational study to assess response rates to the seasonal influenza vaccine in participants with relapsing MS treated with cladribine (ClinicalTrials.gov identifier: NCT05019248). The vaccine-specific antibody responses to the H1N1, H3N2, and B strain 2020/2021 influenza vaccine viruses were measured prior to immunization and six months postvaccination. All included participants had an indication for a seasonal influenza vaccination according to the German national recommendations by the Standing Committee on Vaccination.

Initially, the study was designed to incorporate a larger cohort of 200 patients in order to allow a non-inferiority analysis of the primary endpoint as well as confirmation of various secondary endpoints and immunologic analyses. However, the 2019 coronavirus disease pandemic (COVID-19) and the associated reduction in in-person patient contacts, as well as the limited availability of seasonal influenza vaccine, resulted in slow recruitment, so the originally planned patient count was not achieved. Therefore, we present here results from a smaller study population that were sufficient for a descriptive analysis of the primary outcome and several secondary outcomes (cladribine group: *n* = 90; control group: *n* = 10).

### 2.2. Study Population

All adult patients diagnosed with relapsing MS according to 2017 revised McDonald criteria [16] who underwent treatment with cladribine at the University Hospitals Essen and Duesseldorf, Germany, and who chose to receive a seasonal influenza vaccine on a routine basis were offered to participate in this study. Patients were included from September 2020 to March 2021. Administration of cladribine was performed according to national and international guidelines as well as to the most recent summary of product characteristics (cladribine group; *n* = 90).

The control participants (referred to as “platform DMTs”; *n* = 10) were recruited during the same period and were either treatment naïve (*n* = 1) or received injectable or oral DMTs approved for relapsing MS, namely IFN-β (*n* = 3), glatiramer acetate (*n* = 2), dimethyl fumarate (*n* = 2), or teriflunomide (*n* = 2).

Patients were excluded if they had (i) prior treatment with B-cell-targeted therapies, lymphocyte-trafficking blockers, alemtuzumab, cyclophosphamide, mitoxantrone, azathioprine, mycophenolate mofetil, cyclosporine, methotrexate, total body irradiation, or bone marrow transplantation; (ii) immunosuppressive treatment for diseases other than MS or long-term corticosteroid treatment; (iii) systemic high dose corticosteroid therapy or apheresis procedures 6 weeks prior to vaccination; (iv) contraindications against vaccination.

### 2.3. Vaccination

We initially planned to recruit only patients vaccinated with a single dose of the tetravalent inactivated unadjuvanted split influenza virus vaccine that contained A/Guangdong-Maonan/SWL1536/2019 (H1N1) pdm09 (H1N1GM19), A/Hong Kong/2671/2019 (H3N2, H3N2HK19), B/Washington/02/2019 (BWAS19), and B/Phuket/3073/2013, as recommended for the northern hemisphere by the World Health Organization for 2020/2021, according to the manufacturer’s specification (in detail, the following vaccines were used: Influsplit^®^, Flucelvax^®^, Influvac^®^, and Vaxigrip^®^). However, due to the COVID-19 pandemic, the demand for influenza vaccination increased, resulting in a shortage of the seasonal vaccine for 2020/21. Consequently, several patients had to be vaccinated with a single dose of the trivalent influenza virus vaccine. Therefore, the evaluation was reduced to the immune response against the following influenza strains: H1N1GM19, H3N2HK19, and BWAS19.

In order to evaluate whether the immune response mounted to antigenic stimulation depends on the duration and timing of cladribine therapy, patients were classified into 5 cohorts: those vaccinated shortly (at least 4 weeks) before initiation of cladribine (−3 to −1 months to baseline, cohort 1), early after first cladribine admission (+1 to +6 months to baseline, cohort 2), at the end of the first-year treatment course (+6 to +11 to baseline, cohort 3), shortly after the second treatment course (+13 to +18 months to baseline, cohort 4), or after completing the second year of treatment (>24 months to baseline and >12 months to last cladribine admission, cohort 5). Controls received influenza immunization during treatment with a platform DMT. By implementing this control group, we took advantage of eliminating bias due to disease-specific dysregulations in peripheral immune responses. Based on previous studies, treatment with platform DMTs does not impact the vaccination responses to influenza [8,9,10,11,12,13].

### 2.4. Study Endpoints

The primary endpoint was the proportion of patients with a positive response to the influenza vaccine measured six months after vaccination. A positive response to the vaccine was defined as receiving seroprotection (specific hemagglutination-inhibition (HI) titers ≥ 1:40). As secondary endpoints, we assessed geometric mean antibody titers (GMTs) prior to and six months postvaccination, as well as the proportion of patients with seroconversion (i.e., a prevaccination antibody titer ≤ 10 and a postvaccination HI titer ≥ 40). Finally, adverse events were monitored and collected for all subjects throughout the duration of the study. The severity of adverse events was graded according to Common Terminology Criteria for Adverse Events (CTCAE) version 5.0.

### 2.5. Blood Sampling and Processing

Quantitative antibody titer responses to seasonal influenza vaccines were measured by hemagglutination inhibition assays (HAI). The HIA was applied as described before [9,15,17]. Serum samples for analyses were drawn directly before vaccination (i.e., on the day of vaccination) and six months after vaccination. Five prevaccination samples—all in the cladribine group—could not be processed. However, postvaccination samples were obtained and analyzed from all patients in both groups. Samples were stored at −80 °C until use in the blinded analyses.

In brief, a two-fold dilution series of sera was prepared in phosphate-buffered saline (initial dilution 1:20) and incubated with four hemagglutinin units of whole inactivated H1N1GM19, H3N2HK19, or BWAS19 virus for 1 h. Prior to reading, 1% turkey erythrocytes were added for 1 h. For H1N1GM19 and H3N2HK19 viruses, all tests were performed with positive ferret sera. For BWAS19, sheep serum was used as a positive control. All sera were tested in duplicate. Serum HI titer was expressed as the reciprocal of the highest dilution at which hemagglutination was 50% inhibited [9,17].

The lymphocyte cell counts were assessed via standard hematology laboratory measures. Lymphocyte subsets were assessed in a central laboratory of the University Medicine Essen (CD19+ B cells, CD3+ T cells) using flow cytometry. The total amount of primary immunoglobulin classes (IgG and IgM) was analyzed via latex-enhanced assay by kinetic nephelometry according to manufacturer guidelines [18]. Data were presented as absolute numbers (cells/µL) or unit volumes (immunoglobulins, g/L).

### 2.6. Statistical Analyses

Baseline epidemiologic characteristics were evaluated using descriptive statistics. Comparisons among patient subgroups were made using the χ^2^-test or Fisher’s exact test for categorical variables or the Mann–Whitney rank sum test or Kruskal–Wallis test for continuous variables. All analyses of vaccine response were summarized using descriptive statistics, including the number of patients, mean, and 95% confidence intervals. Unless otherwise stated, the calculation of proportions was based on the number of patients in the analysis set of interest.

## 3. Results

### 3.1. Patients

In total, 90 cladribine-treated patients and 10 control individuals were included in this study and vaccinated against influenza. Except for five prevaccination samples in the cladribine group, complete blood samples (samples before and 6 months after vaccination, respectively) were available from all patients. Demographic characteristics at the time of vaccination between cladribine-treated patients and the control group were generally well-balanced (Table 1). However, 40% of patients in the cladribine group had not received prior DMT, compared to only 10% in the control group. The other patients were previously treated with different substances including IFN-β, glatiramer acetate, and dimethyl fumarate. Our cladribine patients had a median age of 41 (interquartile range (IQR): 31–52) years and a median disease course of 78.5 (IQR: 36.5–180.3) months since MS onset. The median EDSS score was 2.5 (IQR: 1.5–4.0) indicating a moderate disability burden. A comparison of the individual cladribine cohorts showed a significant difference in the annualized relapse rate at vaccination (Table 1). Here, cohorts 1 and 2 demonstrated higher disease activity. Similarly, patients in cohort 2 were younger at the median than the median of the overall cohort (34 (IQR:29–52) vs. 41 (IQR: 31–52)). This is likely due to two factors: (i) only patients with a pronounced active disease course were newly started on cladribine therapy in 2020 since, due to the onset of the COVID-19 pandemic and the associated uncertainty regarding the risk of infection and potentially worse disease outcomes under escalating DMTs (especially in older patients), indications for adjustments were more hesitant; (ii) insufficient time needed for the stabilization of disease activity under cladribine therapy.

### 3.2. Response to Influenza Vaccine

Prevaccination GMTs (HI units) for the influenza strains were comparable between the overall cladribine and control group (H1N1GM19: 80.2 (±95% confidence interval: 46.0–141.3) vs. 76.4 (35.5–164.3); H3N2HK19: 172.2 (83.0–374.1) vs. 252.4 (87.1–731.1); BWAS19: 16.8 (11.5–24.7) vs. 15.2 (9.0–25.9), Table 2). Postvaccination, mean increases in the titers were lower in the cladribine group vs. the control group (H1N1GM19: +98.5 vs. +188.1; H3N2HK19: +225.3 vs. +300.0; BWAS19: +40.0 vs. +78.4); however, titers increased in both groups for all strains six months after vaccination. Comparison between the individual cladribine cohorts showed that increases in GMT levels postvaccination were highest when vaccination preceded cladribine initiation (cohort 1). In particular, the mean titer increase in influenza A strains was lower during the first two years of treatment compared to cohort 1 (H1N1GM19: cohort 1 vs. cohort 2 vs. cohort 3 vs. cohort 4: +153 vs. +70.3 vs. +32 vs. +83.9, respectively; H3N2HK19: +279.1 vs. +157.1 vs. +197.8 vs. +133.9, respectively).

The majority of cladribine patients already had seroprotective antibody titers before vaccination (H1N1GM19: 78.8%, H3N2HK19: 84.7%, BWAS19: 18.8%). Postvaccination seroprotective titers were maintained in those patients. In cladribine recipients, seroprotection rates to all strains were higher after vaccination than before vaccination (H1N1GM19: 94.4%; H3N2HK19: 92.2%; BWAS19: 61.1%). Except for the influenza B strain, postvaccination seroprotection rates were comparable with those of the control group (H1N1GM19: 100%; H3N2HK19: 90.0%; BWAS19: 80%). Seroconversion rates were also comparable between cladribine-treated patients and controls (H1N1GM19: +15.6% vs. 20.0%; H3N2HK19: +7.5% vs. 0%; BWAS19: +42.3% vs. 50%, respectively). However, the interpretability of seroconversion was limited by the small number of patients with seronegative prevaccination titers (HI titer ≤ 10) in both groups.

Concerning the timing of influenza vaccination in relation to the treatment onset, we further observed only mild variation in seroprotection rates, indicating that an effective humoral response can be mounted independently of the duration of cladribine treatment as well as the time interval to the last cladribine administration (Figure 1).

### 3.3. Predictors of Response

We also explored whether the serological response was impacted by lymphocyte counts measured at the time of vaccination. Total lymphocyte counts in patients vaccinated shortly before cladribine initiation were mostly within the normal range (Figure 2). In contrast, patients vaccinated within the first six months after a course of cladribine, and especially within the second year of treatment, typically showed grade I or II lymphopenia. None of the patients in our cohort had severe lymphopenia at the time of vaccination. Notably, most patients maintained or achieved seroprotection independently of total lymphocyte count or subset distribution (Figure 2).

Scatter plots indicate patients with seroprotection following vaccination (black boxes) and without sufficient response to at least one strain (red boxes). Error bars show the 95% confidence interval. BL: baseline; CLD: cladribine; IgG: immunoglobulin G; IgM: immunoglobulin M.

### 3.4. Safety during the Immunization Period

There were no deaths, serious adverse events, or adverse events leading to study discontinuation in either group postimmunization. Measures of cladribine safety during the six months postimmunization were consistent with the phase III safety profile in patients from the clinical development program [19,20], and no new safety signals were identified. Most of the cladribine recipients (51 out of 90, 56.7%) experienced at least one adverse event. Lymphopenia was the most frequently observed event in 34 patients (37.8%).

Except for four cases of grade 3 lymphopenia during the study duration, all adverse events were of mild or moderate intensity. A total of five infectious events occurred in four cladribine patients (4.4%, three cases of upper respiratory tract infection, one case of urinary tract infection, and one case of herpes infection), all infections were rated as mild or moderate. No case of influenza occurred. Two patients (20%) in the control group experienced infectious events, all were mild or moderate in intensity. Similar to the cladribine group, there were no cases of influenza in the control group.

## 4. Discussion

The data reported here demonstrate that patients with relapsing MS treated with cladribine can mount adequate humoral responses to inactivated influenza vaccine.

The humoral response to influenza vaccines depends, among other factors, on the immunogenicity of the strains included. Here, subjects received locally available trivalent or quadrivalent seasonal influenza vaccines (2020/2021). Due to the COVID-19 pandemic and the associated risk of severe complications of co-circulation of influenza and SARS-CoV-2 viruses, the World Health Organization pointed to the 2020–2021 anti-influenza campaign as being of particular relevance. Consequently, vaccines were rare and many patients in our cohort received the trivalent vaccine. Hence, analysis of the fourth strain (influenza B, Phuket) was omitted.

While an increase in GMTs against all influenza strains was lower in cladribine-treated MS patients than in the control group, titers increased in both groups for all strains 6 months after vaccination. Of note, lower GMTs compared with healthy controls or patients on platform DMTs are not specific to cladribine therapy but have also been described for other highly active agents such as fingolimod, natalizumab, and B-cell depleting therapies [9,21,22]. Furthermore, seroprotection rates increased in cladribine recipients for all strains tested, and except for the influenza B strain, seroprotection rates after vaccination were comparable to those in the control group. Thus, the cladribine patients were able not only to maintain their pre-existing specific humoral immunity to influenza but also to mount an immune response anew under therapy. However, the interpretation of per-protocol seroconversion that included an HI titer < 10 (i.e., seronegativity) was challenged by the small number of patients who met this criterion.

Currently, there are few studies on the safety and efficacy of vaccines in MS patients treated with cladribine [15,23,24]. The data available to date relate solely to COVID-19 immunization and few cases of vaccination against seasonal influenza (*n* = 12) and herpes zoster (*n* = 31) during the 2-year prospective phase IV study MAGNIFY [15]. Interestingly, our observations on influenza immunization support previous findings on this vaccine, as well as on the other vaccines mentioned above. All studies noted that patients receiving cladribine were shown to develop specific humoral immunity and the vaccination was considered safe.

In this study, we chose to have a control group under therapy with a platform DMT. By implementing this control group, we take advantage of eliminating bias due to disease-specific dysregulations in peripheral immune responses. Based on previous studies, treatment with basic DMTs does not impact the vaccination responses to influenza [8,9,10,11,12,13]. Nevertheless, a compression to healthy individuals might be interesting and can be the subject of further studies. Data from the literature that allow a comparison with our cohort (e.g., same viral strains, same demographic structure, same time to the outcome, same outcome parameters) do not currently exist. However, it can be said from the literature that in the 2020/21 season the immunological response to H1N1 and H2N3 was higher than to the influenza B strains, even in healthy populations [25].

Whereas the anti-inflammatory activity of cladribine has been attributed to the depletion of memory B cells [26], seroconversion and protection after vaccination upon vaccination in cladribine-treated MS patients are likely the result of immature/naïve B-cell repopulation, occurring after the development of a 1 to 3% B-cell repopulation [27]. In our previous study, we showed that cladribine-treated patients maintained 1% B cell levels and CD19+ B cells recovered to at least 10–20 CD19+ B cells/μL rapidly after cladribine dosing [5]. Therefore, the selective kinetics of lymphocyte repopulation induced by cladribine, including incomplete reduction and subsequent prompt recovery of immature/naïve B cells [26], may explain why vaccine responses in cladribine-treated MS patients resembled those treated with platform DMTs. In contrast, humoral responses to the influenza vaccine are blunted in patients treated with other active DMTs with different mechanisms of action [9,11,14,28].

Under this assumption, it is not surprising that vaccination elicited an effective immune response in all cohorts and that no clear relationship between absolute CD19+ B-cell counts at the time of vaccination and seroprotection rates could be demonstrated, given the depletion and repopulation dynamics mentioned before (with B-cell counts > 1 to 3%).

In addition, it should be noted that sex differences in response to vaccines and especially with regard to the influenza vaccination have been described previously in the literature [29,30,31]. In our cohort, however, we did not find any differences in immune response or adverse events regarding male and female participants. However, the previously reported sex-specific immunologic differences once again highlight the importance of having a balance of women and men in subgroup analyses of vaccine trials.

Moreover, the effect of cladribine treatment on vaccine response in patients with active secondary progressive MS (which is generally covered by the cladribine label in Germany) was not evaluated in this study. While the MS disease course itself would not be expected to affect vaccine responses, age-related decline in immunity is known to impair antibody responses to vaccines in older adults [32]. Since patients with secondary progressive MS tend to be older, they might experience a reduced vaccine response to cladribine.

In addition, the durability of responses to influenza vaccine during treatment is unknown and should be the subject of future studies. Based on our study design with a long follow-up time point (6 months) and the analyzed sub-cohorts of cladribine treatment, it can be expected that the antibody response upon influenza vaccination lasts at least for one influenza season, as demonstrated for patients under platform DMTs [21].

Our study, of course, has some general limitations. Although this analysis was performed prospectively, preconceived patient numbers were not reached, so an indicated non-inferiority approach was statistically not possible. Further, we divided the patient group into smaller groups, also reducing the power of the study. Moreover, our control group was considerably smaller than the cladribine group. In addition, the vaccination response depends on various aspects, such as the immunogenicity of the influenza vaccine, including vaccine factors, adjuvants, individual factors, and repeated vaccination. In our study, patients were immunized with different vaccines. Although all vaccines used contained the same virus strains (tri- or tetravalent), they did so in different doses and with different adjuvants. This further increases the degree of variability but, as mentioned above, was not feasible otherwise due to the vaccine shortage in the acute pandemic. Moreover, the proportion of individuals with seroprotection was already high at baseline. This is likely attributable to the fact that since the 2009 swine flu, the pandemic H1N1 virus strain has been circulating worldwide (with the usual variability) and is, therefore, considered one of the reference strains for seasonal influenza vaccines (so-called pdm09-like virus strains). The same applies to H3N2, where a similar strain has already been integrated into the northern hemisphere vaccines in 2017/2018 and 2016/2017. On the other hand, the influenza B strain BWAS19—in line with our results—was not included in earlier vaccines. Moreover, our patient cohort is likely to have undergone a selection bias resulting from recruiting exclusively at tertiary centers. Nonetheless, this is the first large and prospectively conducted cohort study to assess vaccination response to influenza in this patient cohort.

## 5. Conclusions

In summary, seasonal influenza vaccination is effective and safe in patients who received treatment with cladribine in our cohort, regardless of timing after treatment administration or total lymphocyte count. In keeping with the paradigm of “de-risking immunotherapy” [33], it is recommended to complete vaccination requirements before the initiation of cladribine. However, the annual influenza epidemic season requires a regular refresher of vaccination. In this context, we here demonstrate that vaccination, even after cladribine initiation, generates a substantial humoral vaccine response in most MS patients that is comparable to those treated with platform therapies.

## Figures and Tables

**Figure 1 cells-12-01243-f001:**
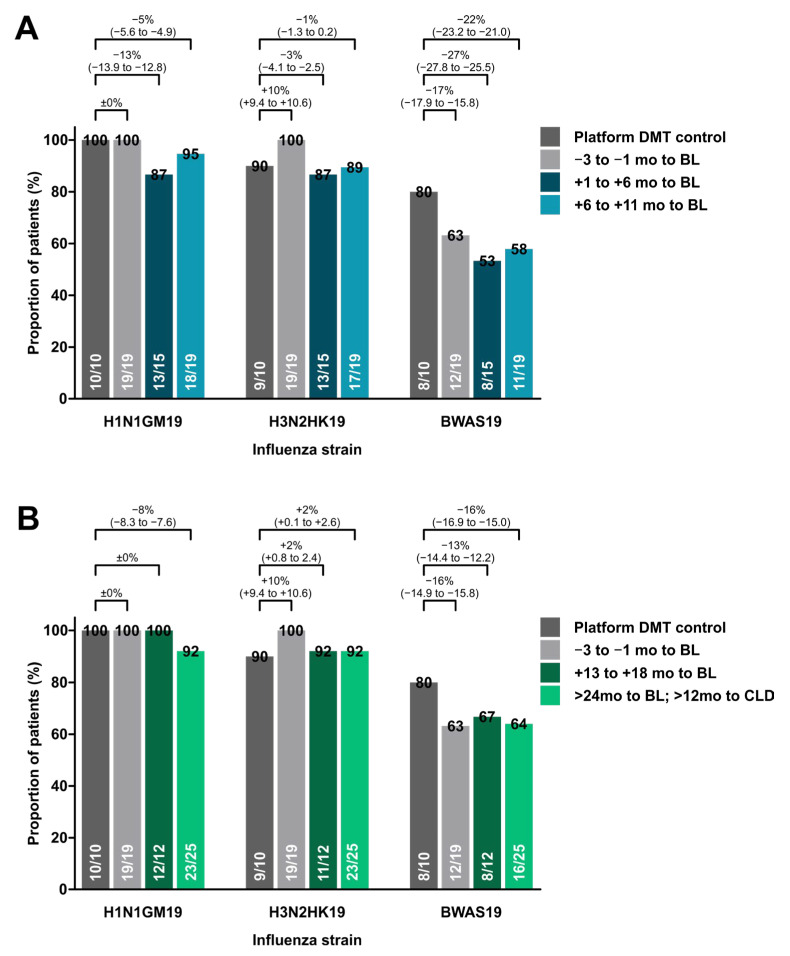
Proportion of patients developing seroprotection to individual influenza strains. (**A**): Proportion of patients with seroprotection among controls and patients having received their vaccination closely to cladribine induction or during year one of treatment. (**B**): Proportion of patients with seroprotection among controls and patients having received their vaccination in year two of cladribine treatment. Data are shown as absolute risk reduction ± 95% confidence interval. DMT: disease-modifying treatment; BL: baseline; CLD: cladribine.

**Figure 2 cells-12-01243-f002:**
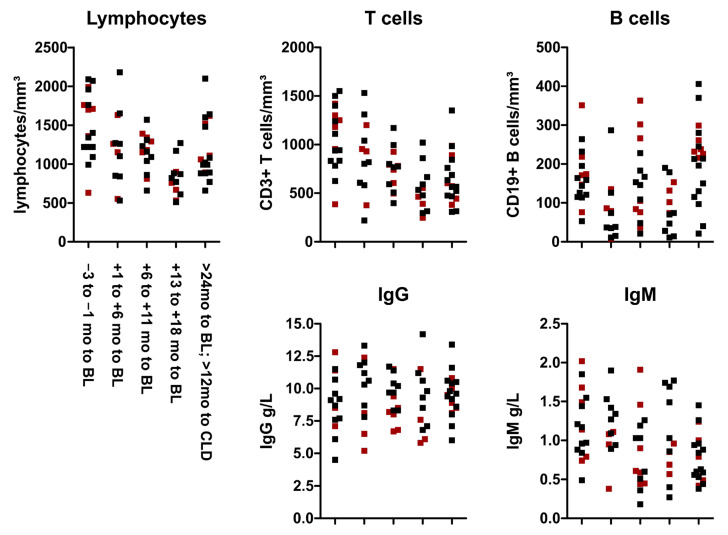
Blood lymphocyte and immunoglobulin levels at vaccination.

**Table 1 cells-12-01243-t001:** Epidemiological parameters at vaccination. Mo: months; BL: baseline; CLD: cladribine; DMT: disease-modifying therapy; yrs: years; No.: numbers; MS: multiple sclerosis; IQR: interquartile range; EDSS: expanded disability status scale; DMF: dimethyl fumarate; TERI: teriflunomide. *: significance level determined using Kruskal–Wallis test; #: significance level determined using Fisher’s exact test.

	−3 to −1 mo to BL (*n* = 19)	+1 to +6 mo to BL (*n* = 15)	+6 to +11 mo to BL (*n* = 19)	+13 to +17 mo to BL (*n* = 12)	>24 mo to BL; >12 mo to CLD (*n* = 25)	Platform DMT Control (*n* = 10)	*p*
Age, yrs, median (IQR)	40 (31–53)	34 (29–52)	41 (35–50)	49 (34–53)	41 (32–51)	53 (31–61)	0.462 *
Male sex, No. (%)	7 (37)	4 (27)	8 (42)	3 (25)	9 (36)	5 (50)	0.797 ^#^
MS duration, mo, median (IQR)	80 (8–160)	13 (7–63)	48 (10–187)	137 (15–231)	68 (25–172)	78 (37–180)	0.075 *
Annualized relapse rate, median (IQR)	1 (1–2)	1 (1–2)	1 (0–1)	1 (0–1)	1 (0–1)	0 (0–0.5)	0.031 *
EDSS, median (IQR)	2.5 (2–4.5)	2 (1–4)	2 (1–3.5)	3.5 (1.5–4.5)	2.5 (2–4)	1.5 (1.5–2)	0.130 *
Number of previous DMT, median (IQR)	1 (0–3)	1 (0–1)	1 (0–2)	1 (0–2)	2 (1–3)	1 (0–1)	0.071 *
Last previous DMT, No. (%)							0.201 ^#^
Naïve	5 (26)	7 (47)	11 (58)	7 (58)	6 (24)	1 (10)	
Platform	5 (26)	2 (13)	2 (11)	1 (8)	8 (32)	5 (50)	
DMF/TERI	7 (37)	6 (40)	4 (21)	3 (25)	8 (32)	4 (40)	
Active	2 (11)	0 (0)	2 (11)	1 (8)	3 (12)	0 (0)	
Vaccine used, No. (%)							0.116 ^#^
Influsplit^®^	9 (47)	3 (20)	9 (47)	6 (50)	16 (64)	7 (70)	
Flucelvax^®^	4 (21)	6 (40)	2 (11)	2 (17)	3 (12)	0 (0)	
Influvac^®^	4 (21)	4 (27)	8 (42)	4 (33)	6 (24)	3 (30)	
Vaxigrip^®^	2 (11)	2 (13)	0 (0)	0 (0)	0 (0)	0 (0)	
Smoker status, No. (%)	8 (42)	7 (47)	5 (26)	2 (17)	11 (44)	3 (30)	0.464 ^#^

**Table 2 cells-12-01243-t002:** Geometric mean titer levels pre- and postvaccination.

**A.**	**Platform DMT Control**		**Overall CLD COHORT**					
**strain**	**pre (*n* = 10)**	**post (*n* = 10)**	**pre (*n* = 85)**	**post (*n* = 90)**				
**H1N1GM19**	76.4 (35.5–164.3)	264.5 (170.5–410.3)	80.2 (46.0–141.3)	178.7 (113.4–284.6)				
**H3N2HK19**	252.4 (87.1–731.1)	552.4 (209.2–1459.0)	172.2 (83.0–374.1)	397.5 (192.5–849.1)				
**BWAS19**	15.2 (9.0–25.9)	93.6 (42.0–208.3)	16.8 (11.5–24.7)	56.8 (31.3–105.2)				
**B**	**Platform DMT Control**		**Cohort 1: −3 to −1 mo to BL**		**Cohort 2: +1 to +6 mo to BL**		**Cohort 3: +6 to +11 mo to BL**	
**strain**	**pre (*n* = 10)**	**post (*n* = 10)**	**pre (*n* = 16)**	**post (*n* = 19)**	**pre (*n* = 14)**	**post (*n* = 15)**	**pre (*n* = 19)**	**post (*n* = 19)**
**H1N1GM19**	76.4 (35.5–164.3)	264.5 (170.5–410.3)	77.1 (45.1–131.9)	230.2 (147.2–360.0)	59.3 (38.4–91.6)	129.6 (63.8–263.3)	78.5 (44.9–137.2)	110.5 (77.8–156.9)
**H3N2HK19**	252.4 (87.1–731.1)	552.4 (209.2–1459.0)	164.7 (92.2–294.3)	443.8 (265.2–742.6)	144.9 (75.2–279.3)	302.0 (127.6–714.9)	149.2 (67.3–330.6)	347.0 (187.0–644.1)
**BWAS19**	15.2 (9.0–25.9)	93.6 (42.0–208.3)	17.7 (11.9–27.0)	54.2 (29.7–98.4)	11.6 (9.1–14.8)	38.9 (19.8–76.3)	17.4 (13.2–22.8)	48.6 (28.5–82.7)
**C**	**Platform DMT Control**		**Cohort 1: −3 to −1 mo to BL**		**Cohort 4: +13 to +17 mo to BL**		**Cohort 5: >24 mo to BL; >12 mo to CLD**	
**strain**	**pre (*n* = 10)**	**post (*n* = 10)**	**pre (*n* = 16)**	**post (*n* = 19)**	**pre (*n* = 11)**	**post (*n* = 12)**	**pre (*n* = 25)**	**post (*n* = 25)**
**H1N1GM19**	76.4 (35.5–164.3)	264.5 (170.5–410.3)	77.1 (45.1–131.9)	230.2 (147.2–360.0)	87.3 (47.1–161.6)	171.2 (105.7–277.2)	102.6 (65.2–161.5)	166.4 (115.4–239.9)
**H3N2HK19**	252.4 (87.1–731.1)	552.4 (209.2–1459.0)	164.7 (92.2–294.3)	443.8 (265.2–742.6)	72.1 (27.4–190.0)	206.0 (78.0–544.1)	249.7 (148.6–419.3)	533.8 (287.9–989.9)
**BWAS19**	15.2 (9.0–25.9)	93.6 (42.0–208.3)	17.7 (11.9–27.0)	54.2 (29.7–98.4)	18.2 (10.9–30.3)	54.6 (34.2–87.3)	20.2 (15.1–27.1)	51.2 (33.5–78.2)

(**A**): Comparison of platform disease-modifying treatment (DMT) controls and the overall cladribine (CLD) cohort subjected to vaccination. (**B**): Comparison of platform DMT controls and patients subjected to vaccination directly prior to CLD treatment (left columns) to vaccination during year one of CLD treatment (right columns). (**C**): Comparison of platform DMT controls and patients subjected to vaccination directly prior to CLD treatment (left columns) to vaccination during year two of CLD treatment and patients having completed year two in the absence of re-treatment or administration of other DMT (right columns). Data are shown as geometric mean ± 95% confidence interval. BL: baseline; mo: months.

## Data Availability

Anonymized data will be shared with qualified investigators upon reasonable request.

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
