# Peer review of "Immune Response to Seasonal Influenza Vaccination in Multiple Sclerosis Patients Receiving Cladribine"

_cells, 2023, doi:10.3390/cells12091243_

Round 1

Reviewer 1 Report

In this paper Rolfes et al studied the immune response in Multiple Sclerosis patients receiving Cladribine, which has immunosuppressive properties. Their results show that patients were able to mount an appropriate immune response to influenza vaccine despite Cladribine treatment. Of note, the seroconversion rate was lower against influenza B patients compared to influenza A. The authors should discuss why that may be the case? Was this related to influenza B not being included in earlier vaccines? The authors acknowledge that the COVID-19 pandemic interfered with recruitment during the trial, however it should also be noted that the number of the patients in the control group (n=10) is much lower than the patients in treatment group (n=90). Finally, different brands of influenza vaccines were used during the trials as can be seen in table 1. This needed to be more clearly mentioned during the methods. The difference between the vaccine’s formulations (adjuvants etc) adds another layer of variability to the study which should also be discussed as a caveat.

In addition to this, please see below some minor comments:

-       How long prior to vaccination was blood samples taken?

-       Line 157, fix typo, “.18”

-       Line 320, fix typo “.5”

-       Line 322, fix type “25”

Author Response

Comments from Reviewer #1:

  1. Of note, the seroconversion rate was lower against influenza B patients compared to influenza A. The authors should discuss why that may be the case? Was this related to influenza B not being included in earlier vaccines?

Response: We thank the reviewer for this important comment and refer to a section from our discussion where this topic is mentioned (lines 375 ff). As the reviewer noted, our guess is that influenza B strain BWAS has not been a part of previous vaccines.

“Moreover, the proportion of individuals with seroprotection was already high at baseline. This is likely attributable to the fact that since the 2009 swine flu, the pandemic H1N1 virus strain has been circulating worldwide (with the usual variability) and is therefore considered one of the reference strains for seasonal influenza vaccines (so-called "pdm09-like" virus strains). The same applies to H3N2, where a similar strain has already been integrated into the Northern Hemisphere vaccines in 2017/2018 and 2016/2017. On the other hand, the influenza B strain BWAS19 - in line with our results - was not included in earlier vaccines.”

  1. The authors acknowledge that the COVID-19 pandemic interfered with recruitment during the trial, however it should also be noted that the number of the patients in the control group (n=10) is much lower than the patients in treatment group (n=90).

Response: We thank the reviewer for this comment and have now included this aspect in the discussion under limitation of the study (lines 366-367).

  1. Finally, different brands of influenza vaccines were used during the trials as can be seen in table 1. This needed to be more clearly mentioned during the methods. The difference between the vaccine’s formulations (adjuvants etc) adds another layer of variability to the study which should also be discussed as a caveat.

Response: We thank the editor for this helpful comment. We have now included this aspect in the method chapter (lines 110-111) and in the discussion (lines 367 ff).

“In addition, the vaccination response depends on various aspects, such as the immunogenicity of the influenza vaccine, including vaccine factors, adjuvants, individual factors, and repeated vaccination. In our study, patients were immunized with different vaccines. Although all vaccines used contained the same virus strains (tri- or tetravalent), they did so in different doses and with different adjuvants. This further increases the degree of variability but, as mentioned above, was not feasible otherwise due to the vaccine shortage in the acute pandemic.”

Minor comments:

  1. How long prior to vaccination was blood samples taken?

Response: We thank the reviewer for this comment and apologize for not being more precise in the first version. Blood samples were collected on the day of vaccination and six months after vaccination. We have now added this missing information in the methods chapter (lines 142-143).

  1. Line 157, fix typo, “.18”

Response: We apologize for the typo and have now corrected it.

  1. Line 320, fix typo “.5”

Response: We apologize for the typo and have now corrected it.

Line 322, fix type “25”

Response: We apologize for the typo and have now corrected it

Reviewer 2 Report

The article by Rolfes et al. presents clinical data regarding seasonal influenza vaccine efficacy in patients on Cladribine therapy for MS compared to typical platform therapy for MS. As noted, the small sample size did prevent certain subcategorical analysis, but the primary objectives of the study were still achieved. I only have minor corrections to recommend.

1.       On lines 320-322, the references are lacking brackets.

2.       It would be easier to see the data distribution in figure 2 if the patients without sufficient response to at least one strain (red boxes), were in separate columns from the seroprotected patients. Error bars or confidence intervals could then be included for the groups.

3.       It would be helpful to include references and discussion comparing the seroprotection and HI titers in healthy individuals to the Cladribine therapy, in addition to the platform therapy controls.

Author Response

Comments from Reviewer #2:

Minor comments:

  1. On lines 320-322, the references are lacking brackets.

Response: We apologize for the typo and have now corrected it.

  1. It would be easier to see the data distribution in figure 2 if the patients without sufficient response to at least one strain (red boxes), were in separate columns from the seroprotected patients. Error bars or confidence intervals could then be included for the groups.

Response: We thank the reviewer for this comment and have now changed figure 2 accordingly.

  1. It would be helpful to include references and discussion comparing the seroprotection and HI titers in healthy individuals to the Cladribine therapy, in addition to the platform therapy controls.

Response: We thank the reviewer for this comment. We attempted to find from the literature a comparison group of healthy individuals with influenza vaccination. However, it was difficult to meet all criteria in this regard: including same vaccination season 2020/2021, same vaccines with same viral strains, same interval to outcome (6 months from vaccination), same outcome parameters (e.g GMT, seroprotection). Unfortunately, we could not find a publication that allows such a comparison. Nevertheless, we have added a paragraph in the discussion where we consider the aspect added by the reviewer among the just mentioned limitations (lines 318 ff).

“In this study, we chose to have a control group under therapy with a platform DMT. By implementing this control group, we take advantage of eliminating bias due to disease-specific dysregulations in peripheral immune responses. Based on previous studies, treatment with basic DMTs does not impact the vaccination responses to influenza .[8-13] Nevertheless a compression to healthy individuals might be interesting, and can be subject further studies. Data from the literature that allow a comparison with our cohort (e.g., same viral strains, same demographic structure, same time to outcome, same outcome parameters) do not currently exist. However, it can be said from the literature that in the 2020/21 season the immunological response to H1N1 and H2N3 was higher than to the influenza B strains, even in healthy populations.[25]”

Reviewer 3 Report

Introduction: No comment.

Materials and methods: 

Line 83-84: The author states that "... we present here results from a smaller study population that were sufficient for a descriptive analysis of  the primary outcome and several secondary outcomes." 

- The information provided does not provide clarity on the size of your cohort.

Results:

- Line 187 -188 : Cladribine patients had a median age of 41 (IQR): 31-52.  Some of the ranges provided in table 1 are out of this IQR. Namely, IQR listed in  column 2 : (31-53), column 3: (29 -52) and column 5: (34 -53). 

- Table 1 provides distribution of the male sex and no further mention in the discussion why this information was important. Or why the information on the female sex was not included.

Discussion: No comment

Author Response

Comments from Reviewer #3:

  1. Line 83-84: The author states that "... we present here results from a smaller study population that were sufficient for a descriptive analysis of the primary outcome and several secondary outcomes." The information provided does not provide clarity on the size of your cohort.

Response: We apologize for not being more precise in the first version. We have now added this missing information on the size of cohorts (lines 83-84).

  1. Line 187 -188 : Cladribine patients had a median age of 41 (IQR): 31-52. Some of the ranges provided in table 1 are out of this IQR. Namely, IQR listed in column 2 : (31-53), column 3: (29 -52) and column 5: (34 -53).

Response: We thank the editor for this comment. Slight differences among the age quartiles have been checked and remain correct. They apparently result from different sample sizes among groups and slight - but ultimately non-significant - skewing of age among groups 1<5. We have now highlighted this in the results section as an example for cohort 2, which shows the largest difference in median from the overall cohort (lines 193 ff.)

  1. Table 1 provides distribution of the male sex and no further mention in the discussion why this information was important. Or why the information on the female sex was not included.

Response: We thank the reviewer for this comment. We have included this information as previous studies have shown that gender may affect influenza vaccine response. We have now included a paragraph on this topic in the discussion chapter (lines 436 ff). In addition, the female gender information in Table 1 is indirectly reported as the difference between total participation per subgroup and male patients.

“In addition, it should be noted that sex differences in response to vaccines and es-pecially with regard to the influenza vaccination have been described previously in the literature.[29-31] In our cohort, however, we did not find any differences in immune response or adverse events regarding male and female participants. However, the previously reported sex-specific immunologic differences once again highlight the importance of having a balance of women and men in subgroup analyses of vaccine trials.”
